Diagnostic value of contrast-enhanced ultrasound and shear-wave elastography for small breast nodules

Shen Yan 1
He Jie 1
Liu Miao 1
Hu Jiaojiao 1
Wan Yonglin 1
Zhang Tingting 1
Ding Jun 2
Dong Jiangnan dongjiangnan0303@126.com 3
Fu Xiaohong fuxiaohong66@126.com 1
1 Department of Medical Ultrasound, Gongli Hospital , Shanghai , China
2 Department of Pathology, Gongli Hospital , Shanghai , China
3 Department of Surgery, Gongli Hospital , Shanghai , China
Soares Paula
Electronic publication date: 2024 Jul 3
Publication date: 2024
Volume: 12
Electronic Location ID: e17677
Received 2023 Oct 9; Accepted 2024 Jun 12
Copyright: ©2024 Shen et al.
Copyright year: 2024
Copyright holder: Shen et al.
License: This is an open access article distributed under the terms of the Creative Commons Attribution License, which permits unrestricted use, distribution, reproduction and adaptation in any medium and for any purpose provided that it is properly attributed. For attribution, the original author(s), title, publication source (PeerJ) and either DOI or URL of the article must be cited.
License URL: https://creativecommons.org/licenses/by/4.0/

Keywords: Contrast-enhanced ultrasound, Shear-wave elastography, Small breast nodule, Ultrasound, BI-RADS classification

Funding: Surface Project of Shanghai Health and Family Planning Commission #202001640 Shanghai Pudong New Area Health System key Specialty Construction #PWZzk2022-18 This study was funded by the Surface Project of Shanghai Health and Family Planning Commission (#202001640) and the Shanghai Pudong New Area Health System key Specialty Construction (#PWZzk2022-18). The funders had no role in study design, data collection and analysis, decision to publish, or preparation of the manuscript.

==============================
Background

The study aims to evaluate the diagnostic efficacy of contrast-enhanced ultrasound (CEUS) and shear-wave elastography (SWE) in detecting small malignant breast nodules in an effort to inform further refinements of the Breast Imaging Reporting and Data System (BI-RADS) classification system.

Methods

This study retrospectively analyzed patients with breast nodules who underwent conventional ultrasound, CEUS, and SWE at Gongli Hospital from November 2015 to December 2019. The inclusion criteria were nodules ≤ 2 cm in diameter with pathological outcomes determined by biopsy, no prior treatments, and solid or predominantly solid nodules. The exclusion criteria included pregnancy or lactation and low-quality images. Imaging features were detailed and classified per BI-RADS. Diagnostic accuracy was assessed using receiver operating characteristic curves.

Results

The study included 302 patients with 305 breast nodules, 113 of which were malignant. The diagnostic accuracy was significantly improved by combining the BI-RADS classification with CEUS and SWE. The combined approach yielded a sensitivity of 88.5%, specificity of 87.0%, positive predictive value of 80.0%, negative predictive value of 92.8%, and accuracy of 87.5% with an area under the curve of 0.877. Notably, 55.8% of BI-RADS 4A nodules were downgraded to BI-RADS 3 and confirmed as benign after pathological examination, suggesting the potential to avoid unnecessary biopsies.

Conclusion

The integrated use of the BI-RADS classification, CEUS, and SWE enhances the accuracy of differentiating benign and malignant small breast nodule, potentially reducing the need for unnecessary biopsies.

Introduction

Breast cancer is the most frequently diagnosed cancer among women and the second leading cause of cancer-related, contributing significantly to morbidity (Siegel et al., 2022; Sung et al., 2021). The asymptomatic detection of breast abnormalities during screenings is prevalent among women diagnosed with breast cancer. In the United States, the 5-year survival rates for localized, regional, and distant breast cancer are 99%, 85%, and 27%, respectively, highlighting the critical importance of early detection and intervention (DeSantis et al., 2017).

Large-scale screening programs have been demonstrated to reduce overall breast cancer mortality by 20% and early-stage mortality by 60% (Nothacker et al., 2009). In China, the higher prevalence of small breast volumes and dense breast tissue represents challenges for mammography, reducing its sensitivity in malignancy detection (Bae & Kim, 2016; Lai & Law, 2015). Ultrasound is crucial for breast cancer screening because of its cost-effectiveness, portability, and accessibility (Geisel, Raghu & Hooley, 2018). In dense breast tissue in particular, ultrasound outperforms mammography in characterizing suspicious lesions (Lee et al., 2010). However, small nodules (≤2 cm) often lack clear ultrasound features, leading to clinical oversight (Welch et al., 2016).

The Breast Imaging Reporting and Data System (BI-RADS) has significantly improved diagnostic accuracy by standardizing breast ultrasonography reporting, thereby increasing the sensitivity in identifying malignant masses (Nam et al., 2016; Zhu et al., 2018). Despite these improvements, the false-positive rate remains high, reflecting the limitations of subjective clinical assessments in conventional ultrasound (Castro et al., 2017; Pistolese et al., 2019; Yeo et al., 2018). This underscores the ongoing necessity for ultrasound-guided biopsies to confirm early breast cancer diagnoses.

According to management guidelines, BI-RADS category 4A nodules warrant biopsies because of their malignancy risk of 3%–10% (Mercado, 2014).

However, biopsies are invasive, they carry needle-related risks, and they impose financial and psychological burdens on patients. Therefore, enhancing screening tools and algorithms to minimize biopsies is imperative.

Advances in contrast-enhanced ultrasound (CEUS) and shear-wave elastography (SWE) offer the potential to refine breast nodule diagnosis. CEUS provides detailed contrast sonograms that highlight the vascularity and morphology of tumors by exploiting tissue-specific acoustic properties (Ji et al., 2017). SWE assesses tissue stiffness by measuring shear wave velocity, aiding in the differentiation of benign and malignant lesions (Youk, Gweon & Son, 2017). Previous investigations indicated that the use of multimodal ultrasound, which integrates CEUS or SWE with traditional ultrasound, can markedly enhance the efficiency of breast cancer diagnosis (Liu et al., 2019; Xiang et al., 2017). Nonetheless, there is a dearth of thorough assessments of the synergistic application of ultrasound, CEUS, and SWE in the detection of small breast cancers and the potential to minimize the number of unwarranted biopsies.

This study evaluated the diagnostic efficacy of CEUS and SWE in detecting small malignant breast nodules to potentially inform further refinements of the BI-RADS classification system.

Materials and Methods

Study design and patients

This retrospective study enrolled all patients with breast nodules who underwent conventional ultrasound, CEUS, and SWE at Gongli Hospital between November 2015 and December 2019. The Ethics Committee of Gongli Hospital approved this study (#[2020] Provisional Trial No. (003)). Given its retrospective design, the requirement for individual consent was waived by the committee. All patient details have been de-identified. The reporting of this study conforms to the STROBE guidelines as per the recommendation (von Elm et al., 2007).

The inclusion criteria encompassed patients who met the following criteria: (1) a breast nodule with a diameter of ≤2 cm with pathological outcomes based on the results of surgical or needle biopsy, (2) no prior treatment for a breast nodule before ultrasonography, and (3) a solid or predominantly solid breast nodule (with the cystic component constituting <25% of the total volume). The breast density of the patients included in this study comprises Type B, Type C, and Type D as determined by ultrasound characteristics. These types reflect a range of breast tissue compositions, including predominantly fibrous (Type B), heterogeneous density with both fibrous and glandular elements (Type C), and predominantly glandular tissue (Type D), which can influence the sensitivity of imaging techniques and the interpretation of findings.

Conversely, the exclusion criteria were (1) pregnancy or lactation and (2) low-quality breast nodule images that were unsuitable for analysis, including cases with blurriness on conventional ultrasound or inconclusive SWE measurements.

For clarity, patients with BI-RADS category 1 lesions, indicating no abnormalities, were excluded from the study. Additionally, patients with BI-RADS category 2 lesions were also excluded. The rationale for this exclusion is that category 2 nodules are typically benign and they often require no further investigation beyond regular follow-up. It is only when these nodules exhibit changes that might warrant a change in the BI-RADS classification that additional imaging or biopsy is considered. A visual representation of the patient selection process is detailed in Fig. 1.

Figure 1 Flowchart of patient selection.

Data collection and imaging examinations

Information on age, tumor size, location, shape, orientation, margins, echo patterns, posterior acoustic features, calcification, vascularity, and lymph node metastasis was retrieved from the patients’ medical records. The multimodality ultrasound examinations were all performed on the same day.

Conventional ultrasound

Operators recorded comprehensive details regarding nodular characteristics, encompassing location, size, shape, orientation, margin, echo pattern, posterior acoustic features, calcification, and vascularity. Subsequently, lesions were classified according to the BI-RADS classification system. In cases in which multiple nodules were present, the nodule with the highest BI-RADS classification was included in the study. If two nodules held the highest BI-RADS classification, both were included concurrently.

SWE

During the study period, a Siemens Acuson S3000 ultrasound diagnostic apparatus (Siemens Medical Solutions, Mountain View, CA, USA) equipped with SWE imaging software was employed for SWE. SWE was conducted when the gray-scale ultrasound indicated that the maximum diameter of the lesion and the image clarity was optimal. The SWE sampling frame size was adjusted to be at least twice the size of the nodules, and patients were instructed to briefly hold their breath after achieving image stability. The SWE speed mode was utilized to directly derive the shear-wave velocity (SWV) across the two-dimensional spatial distribution of the SWE imaging map. The SWV range was gradually fine-tuned (with a maximum of 10.0 m/s) when the interior regions of the nodules displayed red or yellow hues, whereas the surrounding areas appeared blue or green. Multiple regions of interest (typically 5–7) were strategically positioned within various areas within the nodules (upper, lower, middle, and periphery at the highest and lowest speeds). SWV measurements were taken within the effective measurement areas, and the average SWV (m/s) was subsequently determined for each nodule.

CEUS

During the study period, a Philips EPIQ 5 ultrasound diagnostic apparatus equipped with CEUS software (Philips Medical Systems, Bothell, WA, USA) was used. For CEUS, sections exhibiting robust blood flow, prominent blood vessels, or irregular shapes were selected. Areas featuring substantial calcification accompanied by broad sound shadows were intentionally avoided. The focal point was positioned behind the nodule using a mechanical index of 0.07. The contrast agent SonoVue (25 mg in five mL of 0.9% sodium chloride, Bracco SpA, Milan, Italy) was administered per the standard procedure. The nodule’s dynamic perfusion process was observed for at least 3 min. During CEUS, nine distinct variables were assessed (Luo et al., 2016): (1) enhancement intensity (low enhancement, equal enhancement, high enhancement); (2) order of enhancement (concentric, non-concentric); (3) change (difficult to discern, shrinking, unchanged, expanding); (4) enhancement uniformity (uniform, non-uniform); (5) enhancement defects (present or absent); (6) morphology after enhancement (regular, difficult to discern, irregular); (7) enhanced posterior boundary (clearly distinguishable, difficult to discern, unclear); (8) claw sign (present or absent); and (9) presence of nourishing blood vessels. Discrepancies in evaluations were resolved through discussion between two ultrasound physicians to reach a consensus.

Imaging analysis

All SWE image acquisitions and subsequent data analyses were performed by two radiologists with more than 3 and 10 years of expertise, respectively, in SWE and breast ultrasonography. CEUS images were evaluated by two radiologists with more than 5 and 10 years of proficiency, respectively, in CEUS and routine ultrasound examinations. Each ultrasound image was reviewed and confirmed by two ultrasound specialists not involved in the acquisition of the contrast images. The contrast characteristics of the lesions were categorized following consensus agreement. Prior to image analysis, these two ultrasound specialists were not privy to the patients’ clinical data, ensuring an unbiased evaluation process. This approach ensured consistent assessment of CEUS images by the two radiologists. Disagreements between the radiologists were resolved through joint re-evaluation of the imaging features. This collaborative process ensured that any discrepancies were resolved in a manner that maintained the integrity and reliability of the study findings.

To determine the most accurate classification for breast nodules, an integrated multimodal imaging approach was utilized. Initially, each nodule was given a preliminary classification based on the BI-RADS system. Subsequently, nodule vascularity was assessed using CEUS, and any nodule exhibiting at least two malignant features was classified as malignant. The classification was further refined using SWE, with nodules having SWVs of 3.7 m/s or higher being considered malignant. The final categorization was achieved by combining the results of the initial BI-RADS classification with the findings of CEUS and SWE. In cases of disagreement between CEUS and SWE, the BI-RADS classification was either upgraded, downgraded, or retained (Fig. 1). The resulting integrated methodology provided an enhanced diagnostic platform, amalgamating the insights of BI-RADS, CEUS, and SWE. The SWE classification cutoff 3.7 m/s is derived from our team’s clinical experience and a detailed analysis of patient data from the hospital. This value was established through receiver operating characteristic (ROC) curve analysis, which provided the highest accuracy in distinguishing between benign and malignant nodules.

Statistical analysis

The statistical analysis was performed using SPSS 22.0 (IBM, Armonk, NY, USA) and MedCalc 19.0.7 (MedCalc Software bvba, Ostend, Belgium). Continuous variables were reported as means ± standard deviations or ranges, and comparisons were made using the independent-samples t-test. Categorical data were presented as n (%) and analyzed using the chi-squared test or Fisher’s exact test. The agreement between the two radiologists for CEUS and SWE evaluations was determined using the interclass correlation coefficient (ICC), with values close to 1 indicating excellent reliability. To assess the diagnostic efficacy of classifying small breast nodules, ROC curves were employed for the four diagnostic methods (BI-RADS, CEUS, SWE, and the combined method). Areas under the curve (AUCs) were computed to determine the diagnostic performance of the four methods. For statistical evaluation, the Cochran Q-test and z-test were utilized. The optimal cutoffs were derived from the ROC analysis, with subsequent calculation of sensitivity (SEN), specificity (SPE), positive predictive value (PPV), negative predictive value (NPV), and accuracy (ACC). Statistical significance was defined as a two-sided P-value of less than 0.05.

Results

Clinical data and pathological findings

In this study, of the 676 initially assessed patients, 302 met the inclusion criteria, and 305 nodules were analyzed because three patients had two nodules with the same highest BI-RADS classification. Among these, 113 nodules (37.0%) were malignant, and 192 (63.0%) were benign. The mean age of the patients was 49.2 ± 16.4 years. The benign nodules comprised various pathologies, most commonly fibroadenoma (43.2%), adenopathy (21.4%), and adenopathy with fibroadenoma (20.3%). The malignant nodules were mainly invasive ductal carcinoma (69.0%), followed by ductal carcinoma in situ and papillary carcinoma (9.7% each).

Ultrasound predictors of malignancy

Patient age, the breast nodule size, the breast nodule location, and the echo pattern did not exhibit significant differences between patients with benign and malignant nodules (all P > 0.05). However, notable distinctions were observed in terms of shape (P < 0.001), orientation (P < 0.001), margins (P < 0.001), posterior acoustic features (P = 0.001), calcification (P < 0.001), internal vascularity (P < .001), and lymph node metastasis (P < 0.001) between malignant and benign lesions (Table 1 and Fig. 2).

Table 1 The demographic and ultrasonographic characteristics of enrolled patients with small breast nodules.

Parameter	Pathological result	Total	t/χ2	P-value	
	Benign	Malignant				
No. of nodules	n = 192	n = 113	n = 305			
Age, years				0.693	0.406	
Mean	42 ± 14	60 ± 12				
Range	18–83	36–84				
Tumor size (mm)				2.452	0.118	
Mean	13.8 ± 4.3	14.5 ± 3.8				
Range	4–20	5–20				
Location, n (%)				0.759	0.226	
Right breast	88 (45.8)	46 (40.7)	134			
Left breast	104 (54.2)	67 (59.3)	171			
Shape, n (%)				57.455	<0.001	
Oval	106 (55.2)	13 (11.5)	119			
Round	8 (4.2)	7 (6.2)	15			
Irregular	78 (40.6)	93 (82.3)	171			
Orientation, n (%)				59.354	<0.001	
Parallel	159 (82.8)	45 (39.8)	204			
Non-parallel	33 (17.2)	68 (60.2)	101			
Margin, n (%)				91.450	<0.001	
Circumscribed	141 (73.4)	19 (16.8)	160			
Not circumscribed	51 (26.6)	94 (83.2)	145			
Echo pattern, n (%)				3.093	0.542	
Hypoechoic	169 (88.0)	101 (89.4)	270			
Heterogeneous	7 (3.6)	5 (4.4)	12			
Complex cystic and solid	15 (7.8)	7 (6.2)	22			
Hyperechoic	1 (0.5)	0	1			
Posterior acoustic features, n (%)				16.015	0.001	
No. of posterior acoustic features	172 (89.6)	94 (83.2)	266			
Enhancement	10 (5.2)	1 (0.9)	11			
Shadowing	7 (3.6)	17 (15.0)	24			
Combined pattern	3 (1.6)	1 (0.9)	4			
Calcification, n (%)				43.308	<0.001	
None	156 (81.3)	54 (47.8)	210			
Calcification inside a mass	35 (18.2)	52 (46.0)	87			
Calcification outside a mass	1 (0.5)	0	1			
Intraductal calcification	0	7 (6.2)	7			
Vascularity, n (%)				35.871	<0.001	
Absent	136 (70.8)	44 (38.9)	180			
Internal vascularity	43 (22.4)	62 (54.9)	105			
Vessels in rim	11 (5.7)	4 (3.5)	15			
Internal+vessels in rim	2 (1.0)	3 (2.7)	5			
Lymph node metastasis, n (%)				60.745	<0.001	
Normal	192 (100.0)	78 (69.0)	271			
Metastasis	0	35 (31.0)	35			

Figure 2 (A) An irregular and uncircumscribed nodule classified as BI-RADS 4A. (B) CEUS revealed no significant enhancement, and the volume decreased after enhancement. (C) The average SWV on SWE was 2.8 m/s, which was below the cutoff, indicating that the texture of the nodules was soft. After combining BI-RADS, CEUS, and SWE, the BI-RADS category was downgraded to BI-RADS 3. The final pathological result was breast adenopathy. (D) The nodule was not circumscribed and not parallel, and it was classified as BI-RADS 4A. (E) CEUS displayed high enhancement and an increased volume after enhancement. (F) The average SWV on SWE was 4.1 m/s, which exceeded the cutoff, indicating that the nodule was hard.

After combining BI-RADS, CEUS, and SWE, the BI-RADS category was upgraded to BI-RADS 4B. The final pathological result was ductal carcinoma in situ. BI-RADS, Breast Imaging Reporting, and Data System; SWV, shear-wave velocity; SWE, shear-wave elastography; CEUS, contrast-enhanced ultrasound.

Diagnostic efficacy

The BI-RADS classifications of the breast nodules are outlined in Table 2. The determined cutoff for the BI-RADS classification system was category 4B. Correspondingly, this yielded SEN, SPE, PPV, NPV, ACC, and AUC of 82.3%, 74.5%, 65.5%, 87.7%, 77.4%, and 0.784, respectively.

Table 2 The BI-RADS categories of breast nodules.

	Total	Benign	Malignant	Malignant rate (%)	
BI-RADS 3	25	25	0	0.0	
BI-RADS 4A	137	120	17	12.4	
BI-RADS 4B	78	45	33	42.3	
BI-RADS 4C	60	2	58	96.7	
BI-RADS 5	5	0	5	100.0	
Notes.

BI-RADS, Breast Imaging Reporting and Data System.

Interrater reliability, as assessed using ICC, indicated excellent agreement between the two radiologists for both CEUS (ICC = 0.91, 95% confidence interval (CI) [0.89–0.93]) and SWE (ICC = 0.89, 95% CI [0.86–0.91]). The distinct characteristics indicative of potential malignancy in breast nodules identified by CEUS are presented in Table 3. CEUS demonstrated the capacity to identify malignant nodules when at least two of the nine suspicious malignant signs were concurrently present. In this scenario, the ensuing SEN, SPE, PPV, NPV, ACC, and AUC were 83.2%, 87.5%, 79.7%, 89.8%, 85.9%, and 0.853, respectively.

Table 3 Details of the nine suspected malignant characteristics of breast nodules by CEUS.

	Total	Benign	Malignant	Malignant rate (%)	
High enhancement	188	108	80	42.6	
Centripetal enhancement	99	28	71	71.7	
Inhomogeneous enhancement	166	82	84	50.6	
Filling defect	25	2	23	92	
Irregular shape after enhancement	164	70	94	57.3	
Volume expansion	72	8	64	88.9	
Unclear boundary after contrast enhancement	134	38	96	71.6	
Crab foot sign	29	1	28	96.6	
Nourishing vessel sign	57	8	49	86.0	
Notes.

CEUS, contrast-enhanced ultrasound.

The diagnostic efficiency of SWE was determined using the mean SWV. Malignant nodules exhibited a mean SWV of 5.2 ± 1.6 m/s, which was significantly higher the value observed for benign nodules (3.1 ± 1.1 m/s, P < 0.001). Employing ROC curve analysis, the optimal cutoff for SWV was 3.7 m/s. Consequently, the resulting SEN, SPE, PPV, NPV, ACC, and AUC were calculated as 86.7%, 82.8%, 74.8%, 91.4%, 84.3%, and 0.848, respectively.

When CEUS or SWE alone was combined with the BI-RADS classification in diagnosing benign nodules, the BI-RADS classification remained unaltered. Nonetheless, the AUC for this combined diagnosis was 0.758, indicating lower performance compared to the individual BI-RADS classification, SWE, and CEUS. Conversely, the combination of CEUS and SWE led to a one-category increase and decrease in the BI-RADS classification for malignant and benign nodules, respectively. This combined diagnostic approach yielded SEN, SPE, PPV, NPV, ACC, and AUC of 88.5%, 87.0%, 80.0%, 92.8%, 87.5%, and 0.877, respectively.

Significant differences were noted among the four diagnostic methods, as indicated by the Cochran Q-test (Cochran’s Q = 19.573, P < 0.01). The AUCs for the diagnostic efficacy of BI-RADS, CEUS, SWE, and the combined method (CEUS + SWE + BI-RADS) were 0.784, 0.853, 0.848, and 0.877, respectively. In the statistical comparison of the combination diagnostic model with BI-RADS, CEUS, and SWE when utilized individually, P-values of <0.001, 0.012, and 0.017, respectively, were obtained. These results suggest that the combination diagnostic approach is significantly more effective than the individual methods (Tables 4–5 and Fig. 3). Implementing this combined diagnostic approach resulted in the reclassification of 55.8% (67/120) of the BI-RADS 4A nodules as BI-RADS 3. These reclassified nodules were subsequently confirmed as benign through pathological assessment, suggesting that unnecessary biopsy procedures could potentially be avoided for such nodules.

Table 4 Comparison of the diagnoses of malignant nodules based on the four examination methods.

Category	Final diagnosis	Total	χ 2	P-value	
	Benign	Malignant				
No. of nodules	n = 192	n = 113	n = 305			
BI-RADS				92.171	<0.001	
Benign	143	20	163			
Malignant	49	93	142			
CEUS				149.838	<0.001	
Benign	168	19	187			
Malignant	24	94	118			
SWE				143.380	<0.001	
Benign	159	15	174			
Malignant	33	98	131			
Combination				167.533	<0.001	
Benign	167	13	180			
Malignant	25	100	125			
Notes.

BI-RADS Breast Imaging-Reporting and Data System

CEUS contrast-enhanced ultrasound

SWE shear-wave elastography

Table 5 Diagnostic efficacy of the four examination methods.

Parameter	SEN (%)	SPE (%)	PPV (%)	NPV (%)	ACC (%)	AUC	95% CI (%)	
BI-RADS	82.3	74.5	65.5	87.7	77.4	0.784	73.3–82.9	
CEUS	83.2	87.5	79.7	89.8	85.9	0.853	80.9–89.1	
SWE	86.7	82.8	74.8	91.4	84.3	0.848	80.2–88.6	
Combination	88.5	87.0	80.0	92.8	87.5	0.877	83.5–91.2	
Notes.

SEN sensitivity

SPE specificity

PPV positive predictive value

NPV negative predictive value

ACC accuracy

AUC area under the receiver operating characteristic curve

CI confidence interval

BI-RADS Breast Imaging Reporting and Data System

CEUS contrast-enhanced ultrasound

SWE shear-wave elastography

Figure 3 Analysis of the receiver operating characteristic curves of the four methods.

Discussion

This study assessed the potential of CEUS and SWE to distinguish between benign and malignant small breast nodules. The findings indicate that CEUS and SWE could serve as supplementary techniques to enhance the accuracy of the BI-RADS classification for small breast nodules. The combination of CEUS, SWE, and BI-RADS has the potential to enhance the identification of small malignant breast nodules and subsequently reduce the necessity for biopsies.

Through the utilization of CEUS, this study revealed distinct characteristics among fibroadenomas, intraductal papillomas, and malignant breast lesions. Malignant breast lesions often lack capsules, and they feature disorganized capillary networks and increased microcirculation on CEUS, distinguishing them from benign lesions such as fibroadenoma and intraductal papilloma (Chen et al., 2023; Wang et al., 2023). Moreover, the peripheries of malignant breast tumors frequently overlap with regions of breast hyperplasia and various stages of precancerous lesions. Adenosis and inflammatory lesions can exhibit malignancy-like features, such as uneven enhancement and irregular shapes, on CEUS, potentially leading to misdiagnosis (Huang et al., 2019).

Benign tumors, such as breast fibroadenomas, possess stroma with abundant loose mucopolysaccharides, contributing to their reduced hardness compared to malignant tumors such as invasive ductal carcinomas, which are characterized by denser and harder stromal structures because of their fibrous tissue constituents (Aouad et al., 2017). The real-time SWE technique offers a relatively straightforward, noninvasive, and objective approach for assessing tissue hardness. Notably, this study consistently demonstrated the strong diagnostic performance of SWE in terms of SEN and SPE for both BI-RADS 4 nodules and small breast tumors, corroborating prior research findings (Ko et al., 2010; Park et al., 2015). Tissue density data obtained by SWE can predict the extent of vascular infiltration, a key determinant of lymph node metastasis (Celebi et al., 2015; Wojcinski et al., 2012).

It combined CEUS, SWE, and BI-RADS to fine-tune the classification of certain breast nodules. This combined approach resulted in enhanced diagnostic accuracy for both benign and malignant breast nodules. Notably, the study demonstrated that using CEUS and SWE to either downgrade (when both indicators are negative) or upgrade (when both indicators are positive) the BI-RADS classification by one category yielded a superior AUC compared to any of the three methods alone. Significantly, this amalgamated method enabled the reclassification of 55.8% of BI-RADS 4A nodules as BI-RADS 3, all of which were subsequently confirmed to be benign upon pathological examination. This underscores that biopsy could have been avoided for 67 of 120 nodules. Among the falsely identified cases, sclerosing adenoses were the most prevalent, displaying irregular morphologies attributable to interstitial fiber hyperplasia often accompanied by inflammation. These instances corresponded with elevated SWVs exceeding the established threshold coupled with variable regions of pronounced contrast enhancement on CEUS. Conversely, small breast carcinomas that escape detection despite exhibiting malignant features were not as conspicuous on two-dimensional ultrasound, merely displaying significant lobulation, uniform echo patterns, and the absence of posterior feature changes or enhancement. These ultrasound representations bore resemblance to benign tumors, with SWE values below the cutoff and CEUS revealing uniform, low-contrast enhancement. Consequently, a comprehensive analysis of multiple images is imperative for nodules with these attributes, necessitating vigilant monitoring. Furthermore, individuals younger than 60 who have an elevated malignancy risk because of familial history should undergo close surveillance, with additional puncture biopsies performed as deemed necessary (Smith, Cokkinides & Brawley, 2012).

The BI-RADS classification system is not exempt from shortcomings, which are particularly evident in cases involving BI-RADS 4 nodules (in which the probability of malignancy ranges from >2% to <95%) (Jørgensen & Gøtzsche, 2009). In the context of this study, of the 120 nodules initially classified as BI-RADS 4A, 67 were subsequently reclassified as BI-RADS category 3 following the integration of CEUS and SWE results. The diagnostic approach proposed in this study holds the potential to avoid unnecessary biopsies, thereby reducing the associated morbidity linked to breast nodule screening.

Combining ultrasonography modalities for assessing BI-RADS 4 breast lesions continues to be of interest, with studies advocating for various combinations to achieve optimal accuracy. As evident from the recent literature, the amalgamation of ultrasound with two-dimensional SWE and CEUS has displayed considerable promise, especially in studies by Chen et al. (2022) and Liu et al. (2019), which reported AUCs of 0.974 and 0.973 respectively (Table 6). Such high AUCs indicate a substantial diagnostic accuracy. The amalgamated study using a unique combination of BI-RADS, CEUS, and SWE revealed a similar trend, although the AUC was slightly lower at 0.877. Nonetheless, combined modalities enhance the diagnostic precision in assessing BI-RADS 4 lesions. This underscores the potential of these combined techniques as pivotal tools in breast lesion assessment, thus informing clinical decision-making and potentially leading to better patient outcomes.

Table 6 Comparative analysis of combined ultrasonography modalities in assessing breast lesions.

Study	Year of publication	No. of lesions	Benign (%)	Malignant (%)	Best combined modality	AUC	
Chen et al. (2022)	2022	104	82 (78.8%)	22 (21.2%)	US+2D-SWE+CEUS	0.974	
Liu et al. (2019)	2019	118	74 (62.7%)	44 (37.3%)	US+SWE+CEUS	0.973	
He et al. (2023)	2023	26	19 (73.1%)	7 (26.9%)	Combination of CEUS and SWE	0.86	
Our study	2023	305	192 (63%)	113 (37%)	BI-RADS combined with CEUS and SWE	0.877	
Notes.

US ultrasonography

2D-SWE 2D shear wave elastography

CEUS contrast enhanced ultrasonography

BI-RADS Breast Imaging-Reporting and Data System

AUC areas under the curves

Several limitations of this study must be acknowledged. First, its retrospective design inherently carries the risk of selection bias, as the study used pre-existing records that might not have captured all relevant variables uniformly (Hall, Kea & Wang, 2019). Second, the study was potentially subject to inherent biases related to the interpretation of imaging results despite efforts to ensure blinded assessments. Additionally, the study was conducted at a single institution, which could limit the generalizability of the findings to other settings with different patient demographics or varying levels of access to imaging technologies. Finally, although the combination of CEUS and SWE led to improved diagnostic accuracy, the cost and availability of these technologies might pose barriers to their widespread adoption, particularly in low-resource settings (Dan et al., 2023). Future prospective studies with larger, more diverse populations and multicenter collaborations are necessary to validate these findings and explore the cost-effectiveness and practical implementation of integrating CEUS and SWE into routine clinical practice.

The findings of this study have significant implications for clinical guidelines and practices, particularly in resource-limited settings in which access to advanced imaging technologies such as CEUS and SWE could be restricted (Dan et al., 2023). The enhanced diagnostic accuracy achieved by combining CEUS and SWE with the BI-RADS classification system suggests a potential paradigm shift in the management of small breast nodules. However, it is crucial to consider the feasibility and accessibility of these technologies in diverse clinical settings. In regions where these advanced imaging modalities are not readily available, alternative strategies must be developed to ensure that patients receive accurate and timely breast cancer diagnoses (Bonsu & Ncama, 2018; Broach et al., 2016). Future studies should focus on adapting these findings to simpler, more widely available diagnostic tools to ensure broad applicability and minimize disparities in breast cancer care.

CONCLUSIONS

The integrated diagnosis using multiple ultrasound techniques (BI-RADS with conventional ultrasound, CEUS, and SWE) displayed an enhanced capability to differentiate benign and malignant breast nodules. CEUS and SWE might effectively serve as supplementary tools to enhance the clarity of the BI-RADS classification for smaller benign nodules, thereby refining the management of BI-RADS categorization for this subset of cases.

Supplemental Information

Supplemental Information 1 STROBE Checklist

Supplemental Information 2 Small breast nodules data

Supplemental Information 3 Raw Data

Abbreviations

ACC accuracy

AUC area under the ROC curve

BI-RADS Breast Imaging Reporting and Data System

CEUS Contrast-enhanced ultrasound

NPV negative predictive value

PPV positive predictive value

ROC receiver operating characteristics

SEN sensitivity

SPE specificity

SWE Shear-wave elastography

SWV shear-wave velocity

Additional Information and Declarations

Competing Interests

Author Contributions

Human Ethics

Data Availability

The authors declare there are no competing interests.

Yan Shen conceived and designed the experiments, performed the experiments, analyzed the data, prepared figures and/or tables, authored or reviewed drafts of the article, and approved the final draft.

Jie He performed the experiments, analyzed the data, prepared figures and/or tables, authored or reviewed drafts of the article, and approved the final draft.

Miao Liu performed the experiments, authored or reviewed drafts of the article, and approved the final draft.

Jiaojiao Hu performed the experiments, authored or reviewed drafts of the article, and approved the final draft.

Yonglin Wan performed the experiments, authored or reviewed drafts of the article, and approved the final draft.

Tingting Zhang performed the experiments, authored or reviewed drafts of the article, and approved the final draft.

Jun Ding performed the experiments, authored or reviewed drafts of the article, and approved the final draft.

Jiangnan Dong performed the experiments, authored or reviewed drafts of the article, and approved the final draft.

Xiaohong Fu conceived and designed the experiments, performed the experiments, authored or reviewed drafts of the article, and approved the final draft.

The following information was supplied relating to ethical approvals (i.e., approving body and any reference numbers):

The Ethics committee of Gongli Hospital approved the study (#[2020] Provisional Trial No. (003)).

The following information was supplied regarding data availability:

Raw data are available in the Supplemental Files.

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
