# Peer review of "Diagnostic value of contrast-enhanced ultrasound and shear-wave elastography for small breast nodules"

_PeerJ, doi:10.7717/peerj.17677_

## Round 0.1 · original submission · Major Revisions

Dear authors,

As you will see, both reviewers found your manuscript interesting, but suggest further improvements.

Please address all the Reviewers' concerns in a point-by-point letter.

Thank you for choosing PeerJ for your submission.

Best regards

Reviewer 1 ·

Basic reporting

The introduction is long and needs to be more concise.
Clear English writing
Literature references, sufficient field background/context provided

Experimental design

The research question well defined, relevant & meaningful. It is stated how research fills an identified knowledge gap.

Validity of the findings

The idea is good and the authors deal with it carefully.

But I want to ask if the included patients have a different breast density i.e. all breasts are glandular or fatty.

As the study is retrospective why do the authors exclude pregnant or lactating patients?

Additional comments

1- Are all the modalities done on the same day or what is the time interval between them?

2- if there is disagreement between the two radiologists what is the next step for agreement?

3 The author further refined our classification using SWE, where nodules with readings of g3.7 m/s were considered malignant. according to any reference ??
4- Did the lesions with BIRADS 1 & 2 are included in the study or excluded ?

Reviewer 2 ·

Basic reporting

1. The manuscript is well-organized. However, there are a few grammatical errors. I recommend a thorough revision of the manuscript.
2. It provides a comprehensive introduction, detailing the significance of breast cancer detection and the potential of contrast-enhanced ultrasound (CEUS) and shear-wave elastography (SWE) in improving diagnostic accuracy for small breast nodules.
3.References are current and relevant, supporting the context and rationale for the study.
4.The article could benefit from more figures and tables

Experimental design

1.The research question is clearly defined, relevant, and fills an identified knowledge gap regarding the diagnostic efficiency of CEUS and SWE for small breast nodules.
2.The methodology is outlined, including patient selection, imaging techniques, and statistical analysis, allowing for replication of the study.
3. Ethical considerations are addressed with approval from the appropriate ethics committee, and the retrospective design is suitable for the research goals.

Validity of the findings

1. This study contributes valuable insights into the diagnostic process for small breast nodules, suggesting that CEUS and SWE could significantly improve patient care by refining BI-RADS classification.
2. The combination of modalities presents a compelling case for potentially reducing unnecessary biopsies, which is crucial for patient welfare and healthcare resource optimization.
3. While the study presents compelling results, it could benefit from a more in-depth discussion of its limitations, including the retrospective design and any inherent biases.

---

## Round 0.2 · Minor Revisions

Dear authors,

Thank you for the improvements introduced in the Ms, however some previous issues need better attention.

1. Consideration should be given to the potential impact of the study on clinical guidelines and practices, especially in settings where access to advanced imaging technologies might be limited.

2. The study could benefit from a more in-depth discussion of its limitations, including the retrospective design and any inherent biases.

Reviewer 1 ·

Basic reporting

no comment'

Experimental design

no comment'

Validity of the findings

no comment'

---

## Round 0.3 · accepted · Accept

Thank you for your additional revisions.
Your Ms is now acceptable to publication. Thank you for choosing PeerJ to publish your work
Best regards